# Appendix

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

**Risk Categories on Reddit**

R1: Harassment, Hate, and Violence (6 Rules)
R4: Security & Unauthorized Access (4 Rules)
R7: Impersonation & Deceptive Practices (4 Rules)
R10: Age & General Eligibility (1 Rule)
R2: Sexual Content & Exploitation (4 Rules)
R5: Illegal Content & Criminal Activity (4 Rules)
R8: Digital Goods & Economic Feature Misuse (10 Rules)
R11: Intellectual Property (1 Rule)
R3: Privacy & Data Protection (2 Rules)
R6: Platform Integrity & Manipulation (7 Rules)
R9: Moderator Conduct (6 Rules)
R12: Legal Process Misuse (3 Rules)

**Risk Categories on X**

R1: Self-Harm and Suicide (6 rules)
R4: Harassment and Abusive Behavior (10 rules)
R7: Violent Speech and Graphic Media (11 rules)
R10: Username, Account, and Asset Trading (3 rules)
R13: Illegal Activities and Illicit Transactions (3 rules)
R2: Child Sexual Exploitation (CSE) (17 rules)
R5: Hateful Conduct (8 rules)
R8: Privacy and Personal Information (11 rules)
R11: Platform Manipulation and Spam (22 rules)
R14: Election Integrity (1 rule)
R3: Extreme Adult Sexual Content (10 rules)
R6: Extremism, and Terrorism (14 rules)
R9: Deceptive Identities (6 rules)
R12: IP and Copyright (12 rules)

**Risk Categories on Instagram**

R1: Protected-Class Harassment (10 rules)
R4: Dangerous Organizations & Extremism (6 rules)
R7: Suicide, Self-Injury & Eating Disorders (6 rules)
R10: Human Trafficking & Exploitation (3 rules)
R13: Fraud, Scams & Deceptive Practices (8 rules)
R16: Privacy & Personal Data (6 rules)
R19: Misinformation & Manipulated Media (5 rules)
R22: Cybersecurity & Malicious Code (4 rules)
R2: General Harassment & Bullying (11 rules)
R5: Child Sexual Exploitation & Abuse (8 rules)
R8: Adult Sexual Content & Nudity (6 rules)
R11: Drugs, Weapons & Regulated Goods (6 rules)
R14: Identity Misrepresentation & Authenticity (5 rules)
R17: Graphic Violence & Gore (5 rules)
R20: Platform Abuse & Enforcement Evasion (5 rules)
R23: Under-Age & Incapacitated Individual Accounts (4 rules)
R3: Violent Threats & Incitement (10 rules)
R6: Non-Consensual Intimacy (5 rules)
R9: Sexual Solicitation & Prostitution (4 rules)
R12: Criminal Activity & Harmful Acts (7 rules)
R15: Coordinated Interference (5 rules)
R18: IP & Brand Integrity (4 rules)
R21: Account Memorialization (4 rules)

**Risk Categories on Discord**

R1: Harassment & Threats (5 rules)
R4: Violent Extremism & Terrorism (3 rules)
R7: Adult Sexual Content & Conduct (5 rules)
R10: Regulated Goods & Dangerous Instructions (4 rules)
R13: Spam & Platform Manipulation (6 rules)
R16: Gambling (3 rules)
R19: Off-Platform Violence & Harm (1 rule)
R2: Hate Speech & Protected Groups (7 rules)
R5: Self-Harm & Suicide (4 rules)
R8: Non-Consensual Intimate Media (5 rules)
R11: Privacy & Doxxing (3 rules)
R14: Impersonation & Ban Evasion (4 rules)
R17: Financial Scams & Cybercrime (8 rules)
R20: Misuse of Support Systems (2 rules)
R3: Violent & Graphic Content (5 rules)
R6: Child Safety & Exploitation (8 rules)
R9: Sexual Solicitation & Prostitution (3 rules)
R12: Health & Civic Misinformation (5 rules)
R15: Intellectual Property (4 rules)
R18: Human Trafficking (4 rules)
R21: General Compliance (2 rules)

**Risk Categories on YouTube**

R1: Hate Speech (10 rules)
R4: Child Safety (14 rules)
R7: Self-Harm & Suicide (15 rules)
R10: Health Misinformation (11 rules)
R13: Spam & Scams (12 rules)
R16: Impersonation & Identity Deception (7 rules)
R2: Harassment & Cyber-bullying (12 rules)
R5: Sexual Content & Nudity (18 rules)
R8: Violent Extremism & Terrorism (16 rules)
R11: Elections & Civic Integrity (9 rules)
R14: Engagement Manipulation (11 rules)
R17: Platform Integrity & Circumvention (5 rules)
R3: Violent & Graphic Content (17 rules)
R6: Dangerous Acts (16 rules)
R9: Regulated Goods (32 rules)
R12: Census Integrity (5 rules)
R15: External Links & URL Safety (14 rules)
R18: Enforcement & Strikes (4 rules)

**Risk Categories on Spotify**

R1: Violence & Graphic Harm (4 rules)
R4: Self-Harm & Suicide (1 rule)
R7: Health Misinformation (5 rules)
R10: Manipulated & Synthetic Media (4 rules)
R13: Fraud, Phishing & Malware (4 rules)
R16: Platform Integrity & Legal Compliance (4 rules)
R2: Hate & Harassment (6 rules)
R5: Child Safety & Sexual Exploitation (5 rules)
R8: Disinformation & Dangerous Narratives (2 rules)
R11: Impersonation (2 rules)
R14: Illegal & Regulated Goods (4 rules)
R3: Terrorism & Violent Extremism (4 rules)
R6: Adult Sexual Content (2 rules)
R9: Election Integrity & Civic Processes (3 rules)
R12: Non-Consensual Intimacy (2 rules)
R15: Intellectual Property (1 rule)

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

R4: Asset Protection & Confidential Info (2 Rules)    R5: Financial Integrity & Recordkeeping (4 Rules)    R6: Conflicts of Interest & Gifts (4 Rules)
R7: Securities Compliance (1 Rule)    R8: External Communications & Representation (1 Rule)    R9: Anti-Retaliation (1 Rule)

**Risk Categories on Amazon**

R1: Harassment & Discrimination (2 Rules)    R2: Workplace Violence & Threats (2 Rules)    R3: Substance Abuse (2 Rules)
R4: Insider Trading & Material Nonpublic Info (2 Rules)    R5: Antitrust & Fair Competition (1 Rules)    R6: Anti-Bribery & Corruption (1 Rules)
R7: Whistleblower Protection & Anti-Retaliation (1 Rule)

**Risk Categories on Apple**

R1: Policy Compliance & Reporting (2 Rules)    R2: Harassment & Workplace Violence (2 Rules)    R3: Substance Use & Fitness for Duty (3 Rules)
R4: Confidential & Proprietary Information (2 Rules)    R5: Intellectual Property & Technology Use (3 Rules)    R6: Business & Contract Integrity (2 Rules)
R7: External Communications & Representation (3 Rule)    R8: Conflicts of Interest & Outside Activities (8 Rules)    R9: Securities & Insider Trading (1 Rule)
R10: Anti-Bribery, Gifts & Corruption (4 Rules)    R11: Fair Competition & Antitrust (1 Rule)    R12: Political & Use of Resources (1 Rule)
R13: Workplace Privacy (1 Rule)

**Risk Categories on ByteDance**

R1: Legal & Regulatory Compliance (4 Rules)    R2: Whistle-blower Protection & Non-Retaliation (1 Rule)    R3: Information Protection & Privacy (1 Rule)
R4: Record Integrity & Fraud Prevention (3 Rules)    R5: Anti-Corruption & Conflicts of Interest (2 Rules)    R6: Harassment & Workplace Conduct (3 Rules)
R7: Substance Use & Impairment (2 Rule)    R8: Protection of Company Assets (1 Rule)    R9: Occupational Health & Safety (4 Rules)
R10: Labor & Human Rights (4 Rules)    R11: Working Hours & Compensation (3 Rules)

**Risk Categories on Google**

R1: Ethics & Code Compliance (5 Rules)    R2: Equal Opportunity & Respectful Workplace (3 Rule)    R3: Safety & Violence Prevention (6 Rule)
R4: Relationships & Conflicts of Interest (1 Rules)    R5: Confidentiality & Information Security (6 Rules)    R6: Company Assets & Resource Use (2 Rules)
R7: Financial Integrity & Recordkeeping (3 Rules)    R8: Contracting & Signature Authority (1 Rule)    R9: Fair Competition & Antitrust (3 Rules)
R10: Securities & Insider Trading (1 Rule)    R11: Anti-Corruption & Government Dealings (1 Rules)

**Risk Categories on IBM**

R1: Harassment, Bullying & Discrimination (4 Rules)    R2: Workplace Violence & Weapons (3 Rules)    R3: Substance Use & Impairment (3 Rules)
R4: Conflicts of Interest (2 Rules)    R5: Anti-Bribery, Gifts & Political Activity (3 Rules)    R6: Info Protection & Fair Competition (4 Rules)
R7: Integrity of Records & Conduct (5 Rules)    R8: Business Commitments & Use of Company Assets (2 Rules)

**Risk Categories on Intel**

R1: Respect, Violence & Abuse Prevention (3 Rules)    R2: Non-Retaliation & Speaking Up (1 Rule)    R3: Protection of Assets (2 Rules)
R4: Insider Trading & Securities Compliance (1 Rule)    R5: Conflicts of Interest (3 Rules)    R6: Integrity in Communications (2 Rules)
R7: Anti-Bribery & Government Relations (3 Rules)    R8: Fair Competition & Antitrust (3 Rules)    R9: Legal & Regulatory Compliance (6 Rules)

**Risk Categories on Meta**

R1: Respectful Workplace Conduct (4 Rules)    R2: Substance Use & Alcohol (2 Rules)    R3: Conflicts of Interest (2 Rules)
R4: Information Security & Data Privacy (4 Rules)    R5: Financial Integrity & Securities Compliance (2 Rules)    R6: Anti-Bribery, and Gifts (3 Rules)
R7: External Communications & Representation (1 Rule)    R8: Trade Compliance (1 Rule)    R9: Platform Integrity & Illicit Use (1 Rules)

**Risk Categories on Microsoft**

R1: Harassment & Discrimination (2 Rules)    R2: Workplace Violence & Threats (1 Rule)    R3: Substance Abuse & Fitness for Duty (1 Rule)
R4: Investigation Integrity & Info Management (1 Rule)

**Risk Categories on NVIDIA**

R1: Forced & Child Labor / Human Trafficking (4 Rules)    R2: Employment Terms & Worker Freedom (9 Rules)    R3: Harassment & Discrimination (9 Rules)
R4: Workplace Violence & Physical Safety (1 Rule)    R5: Retaliation, Cooperation & Reporting (3 Rules)    R6: Privacy & Transparency (3 Rules)