# OpenReview forum: "GuardSet-X: Massive Multi-Domain Safety Policy-Grounded Guardrail Dataset"
_NeurIPS.cc/2025/Datasets_and_Benchmarks_Track — NeurIPS 2025 Datasets and Benchmarks Track poster_

### Official Review · Reviewer_kq6X · 2025-06-29

**Rating:** 5
**Confidence:** 4

**Summary:**

This paper introduces a dataset, called PolyGuard, to evaluate LLM guardrails. The dataset generation pipeline includes two steps: the first step is to collect domain-specific safety policies and build a hierarchical structure of risk categories and safety rules. Then the second step is to prompt LLMs to generate safe and unsafe examples. Based on this dataset, the authors evaluate 19 advanced guardrail models from various organisations and summaries their results and findings.

**Dataset Code Accessibility:**

Yes

**Ethical Considerations:**

No, there are no or only very minor ethics concerns

**Final Justification:**

My questions are clarified by the authors' rebuttal. And all the other reviewers lean towards acceptance. I will keep my original rating to support the acceptance of this paper.

**Limitations Weaknesses:**

Personally, I feel this paper quite insightful and beneficial for future work. I have two questions:

1. In Finding 4, the authors mention that guardrail models performance better on conversational instances than on single-statement/instruction-only instructions. However, in Figure 5, only the results in the Code and Cyber domains are shown. Is this conclusion generalisable to other domains?

2. In this paper, the dataset is only used for evaluating guardrails. I am wondering whether the same data generation pipeline can also be adopted to train guardrails.

**Strengths Contributions:**

The curated dataset is featured as a comprehensive evaluation of guardrails for various domains, policies, risk categories / safety rules. It could be beneficial for researchers when they build their own guardrails or advance the design of these guardrails. The findings in this paper are insightful and provide valuable consideration when deploying the LLM guardrails.  For example, the researchers could consider training guardrails under more balanced datasets, or eliminating the bias of conservation.

---

> ### Author Rebuttal · Authors · 2025-07-31
>
> We sincerely thank the reviewer for the thoughtful feedback and constructive suggestions!
>
>
> > Q1: Validation of Finding 4 on other domains.
>
> Thank you for the valuable suggestion! We evaluate the F1 scores of conversational inputs versus single-request inputs on the Finance (FINRA) and Law (ABA) domains, as shown in Table 1. To ensure reliability, we focus on a set of well-performing guardrail models to reduce noise from underperforming ones. On average, we observe that conversational inputs improve guardrail performance, consistent with our findings in the Code and Cyber domains. However, in the Finance domain, performance slightly degrades and varies significantly across models. We believe this is due to the longer contextual inputs common in our financial conversations, which may increase guardrail difficulty and reduce model precision. We will incorporate these observations into Finding 4 in our revised version.
>
>
> Table 1: **F1 Score Comparison between Single-Request and Conversational Inputs**.
> | | LlamaGuard 2 | LlamaGuard 3 (1B) | LlamaGuard 3 (8B) | LlamaGuard 4 | WildGuard | Aegis Permissive | Aegis Defensive | Bedrock Guardrail | LLM Guard | Average |
> |-------|--------------|-------------------|--------------------|---------------|-----------|-------------------|------------------|--------------------|-----------|--------------|
> |Single-request input on Finance (FINRA) | 0.621        | 0.468             | 0.432              | 0.569         | 0.712     | 0.264             | 0.627            | 0.473              | 0.744     | 0.546       |
> |Conversational input on Finance (FINRA)| 0.598        | 0.545             | 0.401              | 0.398         | 0.663     | 0.477             | 0.665            | 0.510              | 0.599     | 0.540       |
> |Single-request input on Law (ABA)| 0.611        | 0.486             | 0.345              | 0.545         | 0.703     | 0.176             | 0.484            | 0.421              | 0.642     | 0.490       |
> |Conversational input on Law (ABA)| 0.614        | 0.564             | 0.383              | 0.422         | 0.708     | 0.355             | 0.603            | 0.466              | 0.480     | 0.511       |
>
>
>
> > Q2: Discussion on training guardrails with the data.
>
> Yes, we can certainly partition the dataset or use our pipeline to generate additional data for training guardrail models. Our findings also suggest more effective training paradigms:
> (1) Domain Specialization (Finding 1) indicates the importance of training models with consideration of diverse risks, or enabling them to flexibly adapt to customized risk profiles;
> (2) Model Scaling Stagnation (Finding 3) suggests that a moderate- or small-sized LLM backbone may be sufficient for fine-tuning effective guardrails;
> (3) Adversarial Fragility (Finding 5) and Severity-Skewed Robustness (Finding 6) highlight the need to fine-tune models with attack-enhanced adversarial inputs—especially those from lower-severity categories where benign performance is weaker;
> (4) Conservative Bias (Finding 8) suggests reweighting the loss on unsafe examples to better balance safety with over-conservativeness.
>
> Together, these findings offer valuable insights for training guardrail models, and we hope future work will explore training strategies inspired by them.

---

> > ### Comment · Reviewer_kq6X · 2025-08-02
> >
> > Thank you for your clarification. I will keep my rating to support the acceptance of this paper.

---

### Official Review · Reviewer_v3Ty · 2025-06-29

**Rating:** 4
**Confidence:** 3

**Summary:**

This paper introduces POLYGUARD, a massive multi-domain safety policy-grounded guardrail dataset designed to enhance the safety and robustness of large language models (LLMs). POLYGUARD is notable for being grounded explicitly in real-world safety policies across eight critical domains, including social media, finance, law, and cybersecurity. The dataset incorporates diverse interaction formats, such as declarative statements, instructions, questions, and multi-turn conversations, along with specially crafted benign and adversarial instances designed to evaluate and stress-test guardrail models.

**Additional Feedback:**

The paper does not contextualize the work thoroughly concerning recent relevant literature, such as the notable "Operationalizing a Threat Model for Red-Teaming LLMs." (https://openreview.net/forum?id=sSAp8ITBpC) The authors should discuss and cite this paper explicitly, positioning their work within the broader landscape of LLM safety research. They could incorporate this reference in sections discussing adversarial robustness evaluation and the general methodology of red-teaming.


There are some missing fields in the huggingface dataset
```
Responsible AI Fields

rai:dataCollection: Missing
rai:dataBiases: Missing
rai:personalSensitiveInformation: Missing
rai:dataLimitations: Missing
rai:annotatorDemographics: Missing
rai:dataSocialImpact: Missing
```

I appreciate the insights and takeaways regarding domain specialization, the trade-off in evolving guardrail models, and adversarial robustness, which provide valuable guidance for future research and the practical deployment of safety mechanisms in LLMs.

**Dataset Code Accessibility:**

Yes

**Ethical Considerations:**

No, there are no or only very minor ethics concerns

**Final Justification:**

As far as I am aware, this is the most comprehensive guardrail benchmark, which would be a valuable contribution to the community. The clarification about guardrails being overconservative is interesting because I haven't observed that in my experience. Adding more discussion around this would be helpful. Based on the response and reviews from other reviewers, I maintain my borderline accept rating for the paper.

**Limitations Weaknesses:**

- Missing Prior Work: There are several critical references that are lacking, which would help to contextualize the work within the wider framework of AI security.
- Cultural and Regional Bias: POLYGUARD currently lacks diversity in cultural and regional policies, focusing primarily on Western institutions and global platforms. This limitation could impact the generalizability and inclusivity of the dataset.
- Overrefusal Analysis: While the paper talks about "conservative bias", it does not explicitly categorize overrefusal

**Strengths Contributions:**

- Policy-Grounded Approach: POLYGUARD is uniquely grounded in over 150 real-world safety policies, resulting in 400+ risk categories and 1,000+ detailed safety rules. This explicit grounding in policy documents significantly enhances its relevance for practical, real-world scenarios.
- Domain Coverage and Diversity: The dataset spans eight critical domains, addressing both general and domain-specific safety risks. This breadth enables comprehensive evaluation and potential domain-specific model specialization.
- Interaction Format Diversity: Incorporating various interaction formats (declarative statements, user instructions/questions, and conversations) realistically simulates different user interactions, thereby providing a nuanced testing ground for model behavior.
- Benign and Adversarial Examples: The inclusion of detoxification prompting for creating challenging benign examples and advanced adversarial strategies for attack scenarios enhances the robustness evaluation significantly.
- Clear Takeaways and Findings: Comprehensive experiments involving 19 guardrail models yield insightful findings, such as domain specialization, limitations in generalization across risks, trade-offs in model evolution, and adversarial fragility.

---

> ### Author Rebuttal · Authors · 2025-07-31
>
> We sincerely thank the reviewer for the thoughtful feedback and constructive suggestions!
>
> > Q1: Contextualization of our work within the LLM safety/security literature.
>
> Thank you for the valuable suggestion! The related work by Verma et al. [1] provides a comprehensive survey of threat models and attack methods in red teaming **LLMs**. In contrast, PolyGuard focuses on the safety and robustness of **guardrail models**—lightweight, specialized moderation modules deployed on top of LLMs or integrated into LLM-based systems. These guardrails can be efficiently fine-tuned and externally applied to enforce safety constraints, offering an efficient and flexible approach to LLM safety/security.
> This distinction in red teaming objectives—targeting guardrails rather than base LLMs—motivates the need for a dedicated benchmark. To address this gap, PolyGuard introduces the first large-scale, multi-domain, policy-grounded dataset specifically designed to evaluate and stress-test guardrail models.
> We will consolidate the first paragraph of the Introduction to better situate our work in this broader context, incorporating this clarification and citing related work such as [1].
>
> *[1] Verma, Apurv, et al. "Operationalizing a threat model for red-teaming large language models (llms)." TMLR 2025.*
>
> > Q2: Inclusion of more risks from other regions and cultures.
>
> We appreciate this important point. As noted in our Limitations section (Sec. 5), safety risks and policy standards are continually evolving and can vary across regions and cultures. While our current dataset focuses on widely applicable and cross-regional policies, we have designed our policy-based data generation and red teaming framework to be modular and extensible.
> Specifically, our framework supports the integration of safety policies from diverse institutions and jurisdictions. Extending it to capture region-specific risks is straightforward—for example, by simply incorporating additional region-specific policy documents into the generation pipeline.
> We emphasize that one of PolyGuard’s key contributions lies in providing a scalable and customizable framework for safety-critical data generation. Future work can easily build on this foundation to create culturally or regionally tailored guardrail datasets.
>
> > Q3: Clarifications on overrefusal analysis.
>
> We would like to clarify that in our framework, **“overrefusal” refers to the opposite of “conservativeness”** as discussed in Finding 8. Specifically, overrefusal corresponds to **false positives**, where a guardrail model incorrectly blocks or refuses content that is in fact safe.
> In other words, Finding 8 demonstrates that current state-of-the-art guardrail models tend to be overconservative rather than overrefusive, often failing to block truly unsafe content while being less likely to reject safe inputs.
>
>
> > Q4: Missing fields in Huggingface dataset.
>
> Thank you for pointing this out! In accordance with NeurIPS policy, we are unable to modify the dataset during the review period. We will address this issue and include the missing fields in the final release.

---

### Official Review · Reviewer_M8Ty · 2025-07-03

**Ethics Flags:** Data privacy, copyright, and consent
**Rating:** 5
**Confidence:** 4

**Summary:**

This study focuses on the widespread lack of alignment with standardized safety policies and insufficient attention to domain-specific risks in existing guardrail models and evaluation benchmarks. This paper proposes POLYGUARD—the first large-scale, multi-domain guardrail dataset grounded in real-world safety policies—and conduct a systematic evaluation of 19 guardrail models. The experiments yield several insightful findings, and POLYGUARD provides a new data foundation and improvement directions for building more policy-aligned, risk-consistent, and robust guardrail systems.

**Additional Feedback:**

N/A

**Dataset Code Accessibility:**

Yes

**Dataset Code Comments:**

The code and dataset are provided in submission.

**Ethical Comments:**

The dataset proposed in this study covers a large number of security risk types, including privacy risks. It may be necessary to review whether the relevant samples in the dataset have real privacy information leakage issues.

**Ethical Considerations:**

Yes, there are ethics concerns that require attention by the authors

**Final Justification:**

The author's response resolved my concerns about the types of code vulnerabilities and fairness bias issues in the dataset, and the supplementary experiments also explained the advantages and disadvantages of GPT-4o as a guardrail model.

**Limitations Weaknesses:**

1. Since I have conducted some work related to code security, I paid special attention to the construction of the relevant data. I found that the dataset only annotates whether the code contains vulnerabilities, but does not specify the types of these vulnerabilities. This could be a limitation, as it restricts the dataset’s applicability to broader tasks.

2. Most of the guardrail models compared in the study are trained specifically on a limited set of security risks, and these risk categories are relatively general. Therefore, it is expected that their performance would be poor under the fine-grained, domain-specific risk scenarios presented in this paper. However, general-purpose LLMs like GPT-4, when used as guardrail models, may not suffer from this limitation. If provided with corresponding security rules related to the specific risks, such models might be able to make correct judgments. Hence, I would like to see experimental results in the paper about this aspect.

**Question:**
I have some confusion regarding the issue of code bias. Is this considered an important security vulnerability? Or, if there is a bias problem in the generated code, is it also likely that other issues such as violence, sexual content, or similar harmful content might be present as well?

**Strengths Contributions:**

1. The fine-grained and domain-specific risk framework proposed in this paper enables more precise and interpretable red team evaluations, allowing for accurate identification of model safety flaws at the rule level.

2. The evaluation experiments conducted offer insightful findings, notably revealing that “all models exhibit significant variations in F1 scores across different risk categories, with many models showing high variance, exposing their limited domain coverage and inadequate handling of domain-specific safety issues,” thereby motivating the research community to improve guard models in future work.

3. The appendix of this paper provides detailed descriptions of the dataset construction, including various domains and risk categories, which facilitates the research community in conducting experiments using this dataset.

---

> ### Author Rebuttal · Authors · 2025-07-31
>
> We sincerely thank the reviewer for the thoughtful feedback and constructive suggestions!
>
> > Q1: Clarifications on types of vulnerabilities in the code domain.
>
> Thank you for the valuable comment! In the insecure code domain, our dataset is constructed based on the **CWE Top 25** and **OWASP Top 10**, and indeed includes specific vulnerability types, as detailed in Appendix B.4. These include:
> - CWE Top 25 (25 categories), e.g., CSRF, XSS, SQL Injection, SSRF, Buffer Overflow, Use-After-Free
> - OWASP Top 10 (10 categories), e.g., Injection, Broken Access Control, Security Misconfiguration, SSRF
>
> Together, these 35 categories provide broad and comprehensive coverage of real-world vulnerabilities. We will include vulnerability type annotations in the Hugging Face dataset in the final release.
>
> > Q2: Evaluation of general-purpose LLM with given domain-specific risk categories and rules.
>
> Thank you for the insightful question! We evaluate GPT-4o both with and without access to domain-specific risk categories and safety rules. As shown in Table 1, incorporating these rules as context improves performance across domains. However, this improvement comes with a significant inference cost, as GPT-4o must process hundreds of rules per input. Moreover, GPT-4o is substantially larger than lightweight guardrail models. While domain-specific safety knowledge clearly enhances performance, future work could explore knowledge distillation or fine-tuning to embed these rules into specialized guardrail models, enabling more compact and efficient inference.
>
> Table 1. **GPT-4o performance (F1 score) with and without domain-specific risk categories and rules given as input prompts across domains on a subset of PolyGuard dataset**.
> Each column represents a domain, while rows show performance with and without access to domain-specific categories and safety rules.
>
> | Setting                          | Social Media | General Regulation | Finance | Law    | Education | HR     | Code | Cyber |
> |----------------------------------|--------------|---------------------|---------|--------|-----------|--------|--------------|-----------------------|
> | Without Categories/Rules         | 0.772        | 0.690               | 0.890   | 0.726  | 0.7753    | 0.9176 | 0.6956      | 0.6776                |
> | With Categories/Rules            | 0.859        | 0.849               | 0.956   | 0.916  | 0.9417    | 0.9703 | 0.8641       | 0.7249                |
>
>
> > Q3: More clarification on code bias.
>
> Thank you for the question! We consider code bias a significant issue from a fairness and ethical standpoint, especially in high-stakes domains such as hiring and job performance evaluation. When deployed, biased code can lead to discriminatory outcomes, making it a practical concern, as highlighted in prior work [1]. While we currently do not observe LLM-generated code containing explicit harmful content such as violence or sexual material, and thus do not include such risks in our benchmark, these issues may emerge as model capabilities evolve. Our automatic, policy-aligned data generation framework is designed to flexibly accommodate and generate new data instances as such risks arise.
>
> *[1] Huang, Dong, et al. "Bias testing and mitigation in llm-based code generation." ACM Transactions on Software Engineering and Methodology (2024).*

---

### Official Review · Reviewer_85ob · 2025-07-03

**Rating:** 4
**Confidence:** 3

**Summary:**

This paper introduces PolyGuard, a large-scale policy grounded guardial dataset spanning multiple domains. The authors also evaluate 19 guardrail models on the dataset.

**Dataset Code Accessibility:**

Yes

**Dataset Code Comments:**

The dataset is available in huggingface.

**Ethical Considerations:**

No, there are no or only very minor ethics concerns

**Final Justification:**

The authors has addressed my concerns and I will maintain my score.

**Limitations Weaknesses:**

- **Missing details on the dataset:** The proportion of attack-enhanced instances, as well as declarative statements, questions, instructions,
and multi-turn conversation are missing in the paper. In the dataset provided in huggingface, there is no column indicating the above, making it hard to select some samples based on the desired form.

- **Generation quality**. The instances are generated by LLM. How to ensure the generation quality and correctness without the human annotators?

**Strengths Contributions:**

- The dataset has fine-grained hierarchical categories, along with granular safety rules.

- It includes attack enhanced adversarial examples to evaluate the robustness of guardrail models.

- The authors conduct extensive experiments to benchmark SoTA guard models based on the dataset.

---

> ### Author Rebuttal · Authors · 2025-07-31
>
> We sincerely thank the reviewer for the thoughtful feedback and constructive suggestions!
>
> > Q1: More details on the dataset statistics across different data formats.
>
> To capture diverse interaction formats and realistic security threats, our dataset includes a variety of input types: declarative statements, user questions/instructions, multi-turn conversations, and attack-enhanced examples. To support a clearer understanding of the dataset composition, we now provide detailed statistics of these formats across all domains in Table 1. The format distribution reflects real-world usage patterns. For example, in the Social Media domain, declarative statements and user questions dominate, reflecting typical user behavior on such platforms. In contrast, knowledge-intensive domains like Finance, Law, and Code contain more multi-turn conversations, as users often engage in iterative queries to solve complex tasks. We will add the statistics table and clarification in Section 2.2 and Appendix B.
>
> We also appreciate the reviewer’s suggestion to include format attributes in the Hugging Face release. While we are unable to modify the dataset during the review period according to the NeurIPs policy, we plan to add these annotations in the final release.
>
> Table 1. **Distribution of interaction formats across domains in the PolyGuard dataset**.
> Each cell indicates the proportion of samples belonging to a specific interaction type within the given domain.
> | Domain             | Statement | Instruction/Question | Conversation | Attack-Enhanced |
> |--------------------|-----------|----------------------|--------------|-----------------|
> | Social Media       | 43.8%     | 43.8%                | 0.0%         | 12.4%           |
> | General Regulation | 0.0%      | 45.4%                 | 45.4%         | 9.2%            |
> | Finance            | 0.0%      | 46.0%                | 37.7%        | 16.3%           |
> | Law                | 0.0%      | 40.2%                | 33.3%        | 26.4%           |
> | Education          | 96.8%     | 0.0%                 | 0.0%         | 3.2%            |
> | HR                 | 96.8%     | 0.0%                 | 0.0%         | 3.2%            |
> | Code               | 0.0%      | 47.9%                | 47.9%        | 4.2%            |
> | Cyber              | 0.0%      | 42.4%                | 42.4%        | 15.1%           |
>
>
> > Q2: Clarification and discussion on guaranteeing generation quality.
>
> Thank you for raising the important comment about generation quality! We clarify that although LLMs are used to generate the initial examples, **all instances in PolyGuard were carefully inspected and validated by domain experts** to ensure correctness and high label quality. This process included multiple rounds of human review, both during dataset construction and post-benchmarking case analysis. We believe this rigorous validation is more reliable than relying on automatic filtering, especially since many current SoTA guardrail models still struggle to distinguish safe from unsafe instances in our benchmark.
> Second, we believe it is valuable to explore how to achieve high label precision with reduced human effort. To make the pipeline more automated and scalable to other domains, a promising direction is to develop an agent-based framework in which multiple guardrails collaboratively label data, and samples with low inter-guardrail agreement are routed to humans for active verification [1]. We see this as a compelling future direction for building agent-powered guardrail data generation pipelines that balance the trade-off between annotation cost and label quality.
>
> *[1] Mosqueira-Rey, Eduardo, et al. "Human-in-the-loop machine learning: a state of the art." Artificial Intelligence Review 56.4 (2023): 3005–3054.*

---

### Decision · Program_Chairs · 2025-09-18

**Decision:**

Accept (poster)

**Comment:**

This submission creates a guardrail dataset for evaluating guardrail models, which need to identify correctly whether a given query is allowable or not. What sets this dataset apart from previous ones is to focus less on straightforward, harmful queries, which can often be criticized as overly toy and specific, and instead targetting safety policies scraped from the web. These policies represent concrete guidelines that come from real use-cases, a comprehensiveness that all appreciated as the main contribution of this work. Reviewers had no major concerns by the end of the discussion period, and so the paper should be accepted.

While the ethics reviews mainly considered the permissions and Western bias, there is one aspect that was left undiscussed. That is, the paper mentions risk categories that include child exploitation and terrorism. The authors are strongly recommended to take care that their generated data does not cause undue harm for these especially sensitive topics.